

# Comparing ERA-Interim clouds with satellite observations using a simplified satellite simulator

Martin Stengel[1], Cornelia Schlundt[1], Stefan Stapelberg[1], Oliver Sus[1,2], Salomon Eliasson[3], Ulrika Willén[3], and Jan Fokke Meirink[4]

[1]Deutscher Wetterdienst, Offenbach, Germany
[2]European Organisation for the Exploitation of Meteorological Satellites, Darmstadt, Germany
[3]Swedish Meteorological and Hydrological Institute, Norrköping, Sweden
[4]Royal Netherlands Meteorological Institute (KNMI), De Bilt, The Netherlands

**Correspondence:** Martin Stengel (martin.stengel@dwd.de)

**Abstract.** An evaluation of the ERA-Interim clouds using satellite observations is presented. To facilitate such an evaluation in a proper way, a simplified satellite simulator has been developed and applied to six-hourly ERA-Interim reanalysis data covering the period 1982 to 2014. The simulator converts modelled cloud fields, for example those of the ERA-Interim reanalysis, to simulated cloud fields by accounting for specific characteristics of passive imaging satellite sensors such as the Advanced Very High Resolution Radiometer (AVHRR), which form the basis of many long-term observational datasets of cloud properties. It is attempted to keep the simulated cloud fields close to the original modelled cloud fields to allow a quality assessment of the latter based on comparisons of the simulated clouds fields with the observations.

Applying the simulator to ERA-Interim data, this study firstly focuses on spatial distribution and frequency of clouds (total cloud fraction) and on their vertical position, using cloud top pressure to express the cloud fraction of high, mid-level and low clouds. Furthermore, the cloud-top thermodynamic phase is investigated. All comparisons incorporate knowledge of systematic uncertainties in the satellite observations and are further stratified by accounting for the limited sensitivity of the observations to clouds with very low cloud optical thickness (COT).

The comparisons show that ERA-Interim has generally too low cloud fraction - nearly everywhere on the globe except in the polar regions. This underestimation is caused by a lack of mid-level and/or low clouds - for which the comparisons only show a minor sensitivity to cloud optical thickness thresholds applied. The amount of ERA-Interim high clouds, being higher than in the observations, agrees to the observations within their estimated uncertainties. Removing the optically very thin clouds (COT < 0.15) from the model fields improves the agreement to the observations for high cloud fraction locally (e.g. in the Tropics) while for the mid-latitude regions best agreement of high cloud fraction is found when removing all clouds with COT < 1.0. Comparisons of the cloud thermodynamic phase at the cloud top reveals a too high relative ice cloud frequency in ERA-Interim being most pronounced in the higher latitudes. Indications are found that this is due to the suppression of liquid cloud existence for temperatures below -23°C in ERA-Interim.

The application of this simulator facilitates a more effective use of passive satellite observations of clouds in the evaluation of modelled cloudiness, for example in reanalyses.



## 1   Introduction

In the last two decades major progress has been achieved in improving the representation of clouds in regional and global atmospheric models. Nevertheless, clouds are still one of the largest uncertainties in estimating and interpreting the changes of the Earth's energy budget (Boucher et al., 2013). One of the main problems is that many atmospheric processes cannot be resolved by the current models, which operate on spatial scales ranging from tens to hundreds of kilometres. The small-scale processes related to clouds (e.g., cloud formation and glaciation) are implemented by means of parametrizations that connect grid box mean variables to sub-grid processes. Imperfect parametrizations of clouds will have a significant impact on other model variables and, hence, on the modelled climate sensitivity, which contributes the the large spread among present day climate models in this respect (Dufresne and Bony, 2008). Thus, it is evident that cloud modelling needs to be carefully evaluated and improved to increase our confidence in atmospheric model results. One way to further enhance our understanding of cloud processes is to use high-resolution models (cloud-resolving models, CRMs). They are important tools for testing and improving the parametrizations of cloud-controlling processes, such as cumulus convection, turbulent mixing, and aerosol-cloud interaction (Boucher et al., 2013). However, global and long-term simulations are not feasible with CRMs due to computational cost.

Besides climate general circulation models (GCMs), present day numerical weather prediction models (NWP) also rely on parametrizations to describe subgrid cloud processes. NWP models are also used to generate reanalyses as for example the Modern-Era Retrospective Analysis for Research and Applications, Version 2 (MERRA-2; Gelaro et al., 2017) and ERA-Interim (Dee et al., 2011). As reanalyses are on the one hand closely tied to observations through data assimilations cycles, they can be considered to represent the atmospheric state of the past decades very accurately. On the other hand, discontinuities in assimilated observation systems can create inhomogeneities in reanalysis data, which need to be paid attention to in particular in trend studies. Another inherent shortcoming of reanalyses such as MERRA-2 and ERA-Interim are that cloud properties are still exclusively modelled, due to difficulties in assimilating cloud affected satellite radiances or properly assimilating the cloud properties themselves. The parametrizations in NWP are similar if not identical to the ones used in GCMs. Even though reasonable parameterizations have been developed it remains challenging to balance the system regarding simplicity, realism, computational stability and efficiency in NWP, reanalysis and GCM models.

Indispensable for establishing or increasing confidence in atmospheric modelling is the evaluation of the modelled cloud fields using observations. Satellite measurements are the only source for facilitating this on global scales and corresponding model evaluations using satellite-based datasets have been done in the past for example for columnar-integrated water path (e.g., Waliser et al., 2009; Eliasson et al., 2011) and cloud fraction (e.g., Dai et al., 2006; Sun et al., 2015). Using the standard output fields of models, these studies have already helped evaluating models as well as evaluating long-term monitoring of cloudiness, also in consideration of alternative observation systems. However, for most cloud properties such evaluations often remain difficult due to significant differences in the representativeness of modelled clouds compared to clouds obtained from satellite observations. These differences are most significant in horizontal and vertical resolution, temporal sampling as well as deviating definitions of some geophysical quantities. While model horizontal resolutions could theoretically match those of




satellite observations if enough computational power would be available, with current and past satellite missions at hand it is not feasible to describe the complete, three-dimensional atmospheric state at high temporal resolution with high accuracy and at the same time covering multiple decades.

Satellite retrievals use the measured intensity of radiation from a particular area and direction at a particular wavelength and
infer cloud properties by solving an inverse problem. This implies that several assumptions and ancillary data are required in the forward modelling causing limitations in deriving the geophysical quantities. In addition, passive space-borne instruments usually observe only the top layer of a cloud, while active sensors are able to resolve a cloud vertically to some extent but at the expense of a coarse spatial and temporal global coverage. Consequently, it is plausible that a direct comparison of modelled with observed clouds is suboptimal without building a bridge between the two data sources.

So-called satellite simulators give the opportunity to reduce this problem. They aim for simulating space-borne observations from model fields, which can comprise both simulating derived geophysical properties as well as observed raw measurements. For the remaining part of the paper the focus lies on simulating derived geophysical properties. Figure 1 shows the general concept of such a simulator, which basically covers three steps: (1) adjustment of spatial resolution, observation coverage and frequency, thus the model fields are downscaled and sampled according to the characteristics of the satellite observations,
(2) a pseudo retrieval applied to un-averaged model fields mimicking an actual satellite retrieval also accounting for specific limitations, and (3) statistical aggregation to daily or monthly properties which is ideally done in the same way as for the observational datasets. The inputs to a simulator are usually grid box mean profiles (cloud fraction, cloud water content, temperature, etc.) and surface parameters (surface geopotential, land/sea mask, skin temperature, etc.).

The era of satellite simulators for evaluating climate model clouds began with the ISCCP (International Satellite Cloud
Climatology Project) software package in 1999 (Klein and Jakob, 1999). The ISCCP dataset (Rossow and Schiffer, 1999) along with its corresponding simulator have been used in various studies that diagnosed the performance of GCMs. They have shown that common climate models at that time underestimated the total cloud cover, overestimated the frequency of optically thick clouds and underestimated the frequency of mid-level clouds (Kay et al., 2012). Since then, many other simulators for cloud-related instruments followed (e.g. Pincus et al., 2012) contributing to further improve cloud parameterizations in GCMs.
A widely used tool for interfacing models with satellite observations is COSP (CFMIP Observation Simulator Package), which has been developed by the Cloud Feedback Model Intercomparison Project (CFMIP) community (Bodas-Salcedo et al., 2011). COSP provides the framework for simulating observations and datasets of multiple active and passive satellite instruments (e.g., CloudSat, Calipso, ISCCP, MISR, MODIS) and therefore facilitates an apple-to-apple evaluation of clouds, humidity, and precipitation processes in diverse numerical models. However, many simulators include complex procedures such as radiative
transfer simulations. Most of these procedures can be justified when aiming at performing synthetic satellite retrievals. Caveat of this approach is that so many modifications to the initial model fields are applied, that any comparison to observations will be very difficult to interpret. For example, comparisons between simulated and observed COT should give insights in the accuracy of the COT of the modelled cloud fields.

In this study we follow a more conservative approach. We evaluate modelled clouds in ERA-Interim by employing a simpli-
fied satellite simulator, which can also be seen as a light version of a simulator that keeps the modifications to the model fields



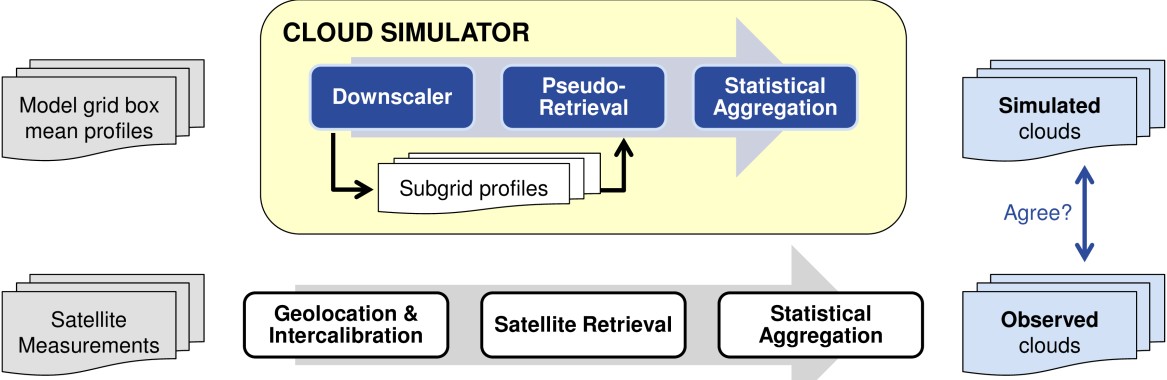

**Figure 1.** Schematic representation of the basic principle of the SIMFERA satellite simulator, which in many aspects is common to other satellite simulators. SIMFERA is a diagnostic tool that maps the model representation of clouds to synthetic satellite observations.

to a minimum. The simplified simulator uses pre-calculated and stored 3-dimensional, instantaneous model fields and accounts firstly for the relatively course spatial resolution of the model compared to observations and secondly for the limitations in the observations in terms of resolving sub-pixel and vertical cloud structures. Cloud variables that are not standard output in the model fields are determined following parametrizations used in the model if available. After undergoing these simplified

simulations, ERA-Interim cloud fields are compared to cloud property observations of the Cloud_cci AVHRR-PM v2.0 dataset (Stengel et al., 2017a) with a focus on systematic climatological deviations between the two sources in the period 1982-2014. Cloud properties addressed are total cloud fraction, the vertical placement of the clouds and the cloud-top thermodynamic phase. These properties have been selected due to the high confidence in them in the observational dataset and because of the availability of quantifying corresponding systematic uncertainties in the Cloud_cci AVHRR-PM dataset by comparison to the

Cloud-Aerosol Lidar with Orthogonal Polarization (CALIOP, Winker et al., 2009) instrument.

     To our knowledge, the application of a satellite simulator to reanalysis data for evaluation of modelled cloud properties has not been published in the peer-review literature yet. Thus, the presented study adds novel aspects to the evaluation of ERA-Interim cloud data, which has generally been very limited so far.

     In this article we will firstly describe the Cloud_cci and ERA-Interim data used in our study (Section 2), followed by a

detailed description of the simplified satellite simulator and a discussion of its output (Section 3). Selected comparisons of cloud fraction, cloud vertical placement and cloud phase are shown and discussed in Section 4. Section 5 summarizes and concludes the study.



## 2 Datasets

### 2.1 The Cloud_cci AVHRR-PM v2.0 dataset

The Cloud_cci AVHRR-PM v2.0 dataset (Stengel et al., 2017a) is a cloud property dataset based on 33 years of AVHRR measurements from the prime AVHRR-carrying afternoon satellites of the NOAA Polar Operational Environmental Satellites

(POES) program: NOAA-07, NOAA-09, NOAA-11, NOAA-14, NOAA-16, NOAA-18, and NOAA-19.

A large variety of cloud properties are included in this dataset, of which the following are used in this study: (1) cloud fraction for all clouds (CFC) and separated into high-, mid- and low-level clouds ($CFC_{high}$, $CFC_{mid}$, $CFC_{low}$), (2) cloud top pressure (CTP) and (3) cloud phase (CPH, here represented by the liquid cloud fraction which is the frequency of liquid clouds with respect to all clouds). The CFC, $CFC_{high}$, $CFC_{mid}$, $CFC_{low}$ and CPH data used are monthly averages, while for CTP

monthly histograms were used, all contained in the Level-3C products defined on a $0.5°$ x $0.5°$ latitude-longitude grid (Stengel et al., 2017b). All available Level-3C products within the time period 1982 through 2014 were used. The underlying, initially retrieved, pixel-level cloud properties were derived using the Community Cloud retrieval for CLimate (CC4CL; Sus et al., 2017; McGarragh et al., 2017).

All remaining details of the Cloud_cci AVHRR-PM v2.0 dataset can be found in Stengel et al. (2017a) also including

validation results for the cloud properties used in this study. All systematic deviations against CALIOP are repeated in Table 1 and extended by the validation scores for optical thickness thresholds ($COT_{th}$) of 1.0 in addition to 0.0 and 0.15.

Cloud_cci cloud detection is successful in 81 % of all pixels being however characterized by an underestimation of cloud occurrence by about 13 %. Removing optically thin clouds from the reference data (CALIOP) clearly improves the agreement of Cloud_cci to the reference with hit rates now reaching 85 % and a bias close to 0. Removing all clouds with COT smaller

than 1 ($COT_{th}$=1.0) from the reference deteriorates the cloud detection scores and leads to a significant positive bias (about 12 % more clouds in Cloud_cci than in CALIOP). This exercise suggests that AVHRR-PM systematically misses the very thin clouds, but builds a very sound reference for all clouds with COT of 0.15 and higher.

Furthermore, cloud top height (CTH) was validated, which is as CTP a representation of the retrieved vertical placement of the cloud top. Similar to the CFC validation COT thresholds were applied, however, here (and for CPH) referring to the optical

thickness into the cloud (top-down) at which the reference value was taken from the CALIOP profile.

Once the liquid phase is correctly identified the CTH is retrieved very accurately with standard deviations below 1 km and biases below 130m. Both measures improve further when removing the very thin cloud layers at the cloud top, although this has only a minor impact on the scores for liquid clouds. For ice clouds the errors in CTH retrieval are larger than for liquid clouds in terms of standard deviations and biases. Removing the thin cloud layers at the cloud top significantly improves the

quality leading to reduced standard deviations of 2.2 km and biases of -0.7 km when selecting the reference CTH at $COT_{th}$=1.0.

For CPH, validation results are very similar to the cloud detection validation, when compared to the reference (CALIOP). Best CPH agreement to CALIOP is found for $COT_{th}$=0.15 into the cloud with hitrate exceeding 80% here, although accompanied with a ice frequency bias of 6.9 %. For $COT_{th}$=0.0 and $COT_{th}$=1.0 hitrate scores are clearly lower.





**Table 1.** Global validation scores of Cloud_cci AVHRR-PM v2.0 cloud mask (CMA), cloud-top height (CTH) and cloud phase (CPH) against CALIOP. For CMA the validation scores are shown as a function of a COT threshold ($COT_{th}$), which is used for separating cloudy from clear-sky CALIOP pixels. For CTH and CPH the scores are also shown as a function of ($COT_{th}$), although ($COT_{th}$) is here referring to the optical thickness into the cloud (top-down) at which the reference value was taken from the CALIOP profile. (Stdd = Standard deviation)

|  | CMA | $CTH_{liq}$ | $CTH_{ice}$ | CPH |
|---|---|---|---|---|
|  | (Hitrate [%]/ Bias [%]) | (Stdd [km] / Bias [km]) | (Stdd [km] / Bias [km]) | (Hitrate [%]/ Bias [%]) |
| $COT_{th} = 0.0$ | 81.2 / -12.6 | 0.91 / -0.13 | 2.84 / -2.65 | 77.1 / 5.9 |
| $COT_{th} = 0.15$ | 84.9 / -0.5 | 0.97 / -0.09 | 2.59 / -1.94 | 80.6 / -6.9 |
| $COT_{th} = 1.0$ | 77.7 / 12.3 | 0.84 / 0.05 | 2.23 / -0.74 | 76.0 / -18.6 |

In summary, comparison with CALIOP have shown that Cloud_cci AVHRR-PM v2.0 data for CMA, CPH and CTH data are of good quality. The detection of optically very thin cloud layers and the retrieval of their cloud phase and cloud top pressure/height using AVHRR is difficult. This however could be well characterized by the presented validation results. Removing the optically very thin clouds from the statistics leads to improvements in the scores (when compared with CALIOP) approach-

ing maximum hitrate scores for CMA and CPH for $COT_{th}$=0.15. For CTH best agreement to CALIOP is found for $COT_{th}$=1.0. In addition to the global validation scores mentioned, latitudinally dependent systematic errors were computed and are shown as uncertainty margin of the Cloud_cci data in the comparisons of section 4.

In addition to these validation studies, the Cloud_cci data was compared to other existing climate datasets such as the Pathfinder extended dataset (PATMOS-x, Heidinger et al., 2014) and the Climate Monitoring Satellite Application Facility's

(CM SAF) cloud, albedo and radiation dataset (CLARA-A2, Karlsson et al., 2016) in Stapelberg et al. (2017), which documented reasonable similarities for the cloud properties considered in this paper.

It is also important to note that an inherent feature of passive imagers is that most of the time only the uppermost cloud can be observed. This has the direct consequence that clouds covered by cloud layers above will not be detected. This limitation however has been accounted for in the simulator presented in section 3.

## 2.2   ERA-Interim reanalysis

The ERA-Interim global atmospheric reanalysis (Dee et al., 2011) provided by the European Centre for Medium-Range Weather Forecasts (ECMWF) is the follow-up of ERA-40 reanalysis (Uppala et al., 2005). ERA-Interim covers the period from 1979 onwards and is continuously extended operationally. One of the main objectives was to solve various difficulties regarding data assimilation (e.g. use of satellite data), which were found during the production of ERA-40. See for example

Dee et al. (2011); Mooney et al. (2011); Bao and Zhang (2013); Betts et al. (2009) for improvements of ERA-Interim over ERA-40.

The ERA-Interim reanalysis is produced by the Integrated Forecast System (IFS) version Cy31r1, which includes the forecast model consisting of three fully coupled components for the atmosphere, land surface, and ocean waves. ERA-Interim clouds are represented by a fully prognostic cloud scheme in which cloud-related processes are treated in a unified way, i.e. they are





physically realistic and consistent with the rest of the model. More specifically, cloud processes are described by prognostic equations for cloud condensate and cloud fraction obeying mass balance equations (Tiedtke, 1993). Clouds are defined by the horizontal coverage of the grid box by cloud and the mass mixing ratio of total cloud condensate, along with the constraint that cloud air is saturated with regard to water and ice, respectively. As time evolves in the model simulation the cloud variables

change due to source and sink terms that are related to cloud formation (e.g., condensation/sublimation, cumulus convection) and destruction (e.g., evaporation, precipitation) processes, respectively. The cloud water is separated into ice and liquid portion by a temperature dependent function between -23 and 0°C constrained by only ice water below and only liquid water above this temperature range (ECMWF, 2007).

As for most large-scale models, the fact that only bulk properties of clouds can be taken into account is an important and

indispensable limitation of ERA-Interim.

ERA-Interim in general has been used in many climate studies in the past (e.g. Screen and Simmonds, 2010; DeMott et al., 2013; Madonna et al., 2014; Simmons and Poli, 2015), including studies on clouds (e.g. Jiang et al., 2011; Cuzzone and Vavrus, 2011; Hanley and Caballero, 2012).

## 3 SIMFERA - a simplified satellite simulator

The SIMplified satellite simulator For ERA-interim (SIMFERA) reads the 6-hourly (00, 06, 12, 18 UTC) gridded, three-dimensional (3D) model fields of meteorological upper air parameters on 60 model levels with the top of the atmosphere located at 0.1 hPa. These model fields include liquid water content $LWC_{gbm}$(the mass of condensate per mass of moist air in [kg/kg]), where the subscripts gbm refers to grid box mean values, ice water content $IWC_{gbm}$ [kg/kg], cloud cover, temperature ($T$) [K], and specific humidity ($Q$) [kg/kg]. Additionally, the data input comprise the surface geopotential ($Z$) [m$^2$/s$^2$] and the

logarithm of surface pressure [Pa] which are used to compute the vertical pressure and geopotential profiles at model levels.

### 3.1 Pre-processing

In the pre-processing step the $LWC_{gbm}$ and $IWC_{gbm}$ at each model level are divided by cloud cover, yielding the so-called in-cloud liquid and ice water content (LWC and IWC) at each model level. For the layers in between two consecutive levels, LWC and IWC are used to determine layer liquid and ice water path ($LWP_{lay}$ and $IWP_{lay}$) incorporating the height of the layer.

The liquid and ice cloud optical thickness per layer ($lCOT_{lay}$, $iCOT_{lay}$) are obtained by rearranging the Han et al. (1994) formulation originally defined for diagnosing LWP from COT and CER:

$$COT = \frac{3}{4}\frac{CWP \cdot Q_{ext}}{CER \cdot \rho} \tag{1}$$

where CWP represents either LWP or IWP depending on the thermodynamic phase. $Q_{ext}$ denotes the extinction coefficient, which is assumed to be 2 for water and 2.1 for ice. The density $\rho$ is set to 1000 kg/m$^3$ for water and 916.7 kg/m$^3$ for ice.

The computation of CER in SIMFERA is done as in the ERA-Interim radiation scheme: (a) following Martin et al. (1994) for liquid clouds where $CER_{liq}$ is a function of liquid water content and cloud droplet number concentration (CDNC). The





needed number of cloud condensation nuclei is 300 cm$^{-3}$ over land and 100 cm$^{-3}$ over sea; (b) for ice clouds, CER$_{ice}$ is a function of temperature and ice water content based on Sun and Rikus (1999) and revised by Sun (2001).

## 3.2 Downscaler

The second part of the simulator addresses firstly the mismatch in horizontal scale between an ERA-Interim model grid box (∼ 80 km) and a satellite footprint (∼ 5 km for AVHRR global data), and secondly, the limitation in vertical resolution of the AVHRR based satellite retrieval. For adjusting the spatial resolution (and for resolving sub-grid variability within a model grid cell), the vertical profiles in each model grid cell are projected onto a certain number of subcolumns, each of which can be thought of as representing an AVHRR pixel. At each model layer, the downscaler distributes the grid-cell cloud fraction randomly onto the subcolumns. Clouds occurring in layers on top of each other (vertically neighbouring layers) are are assumed to overlap maximally, while vertically separated cloud layers are assumed to have a random overlap. This procedure follows the maximum-random overlap approach of (Geleyn and Hollingsworth, 1979). This approach for deriving subgrid profiles is very similar to SCOPS (Subgrid Cloud Overlap Profile Sampler, Webb et al. (2001)), which is implemented, for instance, in COSP.







**Figure 2.** Example case for converting ERA-Interim grid box profiles to SIMFERA subcolumns and to pseudo retrievals. The left column shows ERA-Interim based profiles of cloud fraction (a), cloud phase (b), layer water path (c) and layer cloud optical thickness (d). The middle column gives the same data after columnizing into 20 subcolumns representing the sub-grid variability and after removing the uppermost cloud layers with layer optical thicknesses below a certain threshold (here 0.15): cloud mask (e), cloud phase (f), layer water path (g, layer liquid/ice water path in liquid/ice cells) and layer optical thickness (h, layer liquid/ice optical thickness in liquid/ice cells). The right panels show vertically summarized values per subcolumn (also called pseudo retrieval): cloud mask (i), cloud top phase (j), vertically integrated water path (k), vertically integrated optical thickness (l) and cloud top pressure (m). Each of these subcolumn-representative values can be seen as an individual pseudo satellite retrieval.




Figure 2 visualizes the results of the preprocessing and downscaling for a randomly selected case. To summarize, the pre-processing calculates LWC and IWC from $LWC_{gbm}$ and $IWC_{gbm}$ which are further processed to $LWP_{lay}$ and $IWP_{lay}$ (panel c) (for each of the $M$ model layers), which are in turn used to calculate $lCOT_{lay}$ and $iCOT_{lay}$, liquid and ice optical thickness per layer (panel d). The downscaler creates a $N$-by-$M$ matrix for each model grid cell with $N$ being the number of subcolumns. In the framework of this study, $N = 20$ was used, which on the one hand resembles a subgrid at a courser resolution compared to the resolution of the AVHRR GAC data, on the other hand this number was found to be a good compromise as no significant change in the later on calculated grid-mean values was detectable compared to larger $N$s, while the computational expense increases linearly with increasing $N$. All cells in the $N$-by-$M$ matrix can be either cloudy (1) or not cloudy (0) and the cloudy cells can be of liquid or ice phase. Depending on the phase of each cell the COT matrix is filled with $lCOT_{lay}$ or $iCOT_{lay}$ and the CWP matrix with $LWP_{lay}$ or $IWP_{lay}$. In each subcolumn, all uppermost cloud cells for which the vertically, top-down integrated COT is smaller than the chosen COT threshold are removed. In this study three different COT thresholds were used: $COT_{th}$=0.0, $COT_{th}$=0.15 and $COT_{th}$=1.0. In case the total COT in a subcolumn is smaller then the threshold that subscolumn is counted as clear. It should be noted that applying a threshold of $COT_{th}$=0.0 is a good approximation of not using a simulator at all when considering CFC because no clouds are excluded from the statistics. For cloud top properties, such as CPH and CTH, the application of a simulator is mandatory since these properties are not included in the standard reanalysis output.

### 3.3 Pseudo-retrieval and data aggregation

As the spatial scales of the subcolumn (Figure 2e-h) now represent those of AVHRR pixels, a pseudo-retrieval is done for each subcolumn based on its vertical cloud distribution. CTP, CER and CPH are collected from the uppermost cloud layer. COT and CWP are integrated over the entire column. If a subcolumn does not contain a single cloud cell it is assigned to be clear, otherwise cloudy.

Figure 2 further shows the pseudo retrievals for each subcolumn (panels i-m) for the presented example case. CTP values are additionally converted to cloud top height (CTH) and cloud top temperature (CTT). COT, CER and CWP are not considered during twilight and night-time conditions (sun zenith angles above 75°) to be consistent with the observational data.

All pseudo-retrievals from all subcolumns are used as input for aggregating monthly properties (averages and histograms) on a latitude-longitude grid of 0.5° resolution. A list of monthly mean cloud properties produced by SIMFERA is shown in Table 2. Spatially resolved histograms are composed following the Cloud_cci definition (see Table 5 of Stengel et al. (2017a)) with all histograms being compiled for liquid and ice clouds separately. As CTP histograms are used later in this study, their bin borders are repeated here:

{ 1, 90, 180, 245, 310, 375, 440, 500, 560, 620, 680, 740, 800, 875, 950, 1100 } hPa.

For the comparisons shown and discussed in section 4 SIMFERA was applied to ERA-Interim data from 1982 to 2014.



**Table 2.** SIMFERA output of monthly mean cloud properties. Bold font marks properties used for comparisons to observations in this study. The separation of low, mid-level and high clouds at 680 hPa and 440 hPa for $CFC_{low}$, $CFC_{mid}$ and $CFC_{high}$ follows Rossow and Schiffer (1999).

| Variable name | Abbrev. | Unit | Description |
|---|---|---|---|
| **Total cloud fraction** | **CFC** | 1 | Total cloud fraction (all clouds) |
| **Low level cloud fraction** | **$CFC_{low}$** | 1 | Fraction of low-level clouds with CTP larger than 680 hPa |
| **Mid level cloud fraction** | **$CFC_{mid}$** | 1 | Fraction of mid-level clouds with CTP between 680 hPa and 440 hPa |
| **High level cloud fraction** | **$CFC_{high}$** | 1 | Fraction of high-level clouds with CTP lower than 440 hPa |
| **Liquid cloud fraction** | **CPH** | 1 | Fraction of liquid clouds with respect to all clouds |
| Cloud top pressure | CTP | hPa | Pressure level at uppermost cloud layer top |
| Cloud top height | CTH | km | Derived from CTP and atmospheric profile |
| Cloud top temperature | CTT | K | Derived from CTP and atmospheric profile |
| Cloud effective radius | CER | $\mu$m | Effective particle radius at cloud top |
|  |  |  | (additionally stratified by cloud top phase ($CER_{liq}$,$CER_{ice}$)) |
| Cloud optical thickness | COT | 1 | Vertical integrated cloud optical thickness |
|  |  |  | (additionally stratified by cloud top phase ($COT_{liq}$,$COT_{ice}$)) |
| Cloud liquid water path | LWP | g/m$^2$ | Vertical integrated cloud water path of liquid clouds |
| Cloud ice water path | IWP | g/m$^2$ | Vertical integrated cloud water path of ice clouds |

## 4 Comparison of ERA-Interim and Cloud_cci cloud properties

In this section the cloud properties of ERA-Interim, which have been adjusted by SIMFERA to simulate satellite retrieved properties, are compared against the observed properties of the Cloud_cci AVHRR-PM v2.0 dataset by considering climatologies of CFC, CPH and $CFC_{low}$/$CFC_{mid}$/$CFC_{high}$ mean values and CTP histograms. In addition, monthly mean/histogram fields of both data sources were averaged/aggregated to multi-annual properties within the time period 1982 to 2014 (33 years). ERA-Interim cloud properties are presented threfold by applying three different COT thresholds as described in Section 3.2. For all zonal mean comparisons the estimated systematic uncertainty of Cloud_cci data is given to facilitate a better interpretation of the results.

### 4.1 Cloud fraction

Figure 3 shows multi-annual mean CFC for Cloud_cci AVHRR-PM data and for ERA-Interim. In general, ERA-Interim clouds present very similar global patterns compared to Cloud_cci with high cloud fractions in the mid- and high-latitude storm track regions, in the Artic, in the inner-Tropics and in regions with persistent marine stratocumulus. Low cloud fractions are in particular found in the subtropical subsidence regions. When considering all clouds in ERA-Interim ($COT_{th}$=0.0) a general underestimation of CFC is however found compared to Cloud_cci. This underestimation in ERA-Interim is outside





**Figure 3.** Multi-annual mean cloud fraction (CFC) from ERA-Interim (panels a-c) and Cloud_cci (panel d), where the ERA-Interim cloud fraction was produced by SIMFERA for three optical thickness thresholds (COT$_{th}$=0.0, 0.15, 1.0, respectively). Panel e: zonal mean plot of CFC for all 4 sets. The grey shaded area corresponds to the estimated systematic uncertainty in Cloud_cci CFC based on comparisons to CALIOP. The uncertainty in Cloud_cci is mainly due to missing optically very thin clouds.

the estimated systematic uncertainty of Cloud_cci. Exceptions are the polar regions in which ERA-Interim has significantly higher CFC than Cloud_cci, even outside the reported uncertainty range. The general underestimation of CFC in ERA-Interim outside the polar regions is in line with results found by Free et al. (2016) for the continental US. Removing the optically thin clouds from ERA-Interim, which had been found to be under-represented in Cloud_cci, further increases the underestimation

5   of ERA-Interim CFC compared to Cloud_cci between 60°S and 60°N. The reduction in CFC, when removing the optically thin clouds, is highest in the polar regions, e.g. over Antarctica more than 20 % of all clouds in ERA-Interim have optical thicknesses lower than 0.15. Removing all clouds with an optical thickness below 1 leads to a further reduction in CFC, most





prominently in the high latitudes and polar regions. The agreement between the two datasets and the sensitivity of ERA-Interim to changes in $COT_{th}$ are remarkably different for different cloud levels, which is elaborated on in the next subsection.

## 4.2 Vertical cloud distribution

Figure 4 shows multi-annual mean cloud fraction for three vertical layers: $CFC_{high}$, $CFC_{mid}$ and $CFC_{low}$. In contrast to the

total cloud fraction, for which a significant underestimation was found in ERA-Interim, the high cloud fraction reveals partly opposite characteristics. Many more high clouds are found in ERA-Interim than in Cloud_cci. The difference amounts up to 20 % cloud fraction, being largest in the mid and higher latitudes, but also in the Tropics the difference is approximately 10 %. In the Cloud_cci dataset, $CFC_{high}$ also has the highest systematic uncertainty as again shown by the grey shading embedding the ERA-Interim results. Removing the optically thin clouds from ERA-Interim reduces the $CFC_{high}$ significantly, highlighting

the large contribution of optically thin clouds to ERA-Interim $CFC_{high}$. When applying $COT_{th}$=0.15, ERA-Interim $CFC_{high}$ drops by 5 to 10 % - still being higher than Cloud_cci in the mid and high latitudes. In the Tropics, ERA-Interim $CFC_{high}$ is already lower than Cloud_cci in this set-up. For $COT_{th}$=1.0 ERA-Interim $CFC_{high}$ is more than halved, now being below Cloud_cci for all latitudes.

Considering mid-level clouds, the ERA-Interim cloud fraction is lower than in Cloud_cci. The difference in $CFC_{mid}$ is

about 5 to 10 %, correlating with the amount of $CFC_{mid}$. The systematic uncertainty in Cloud_cci $CFC_{mid}$ has a different sign compared to $CFC_{high}$, which is assumed to be a results of high clouds being classified as mid-level clouds, as the CTP retrieval in Cloud_cci is biased high for these thin clouds. The ERA-Interim $CFC_{mid}$ has nearly no sensitivity to the varying $COT_{th}$ (except over Antarctica), showing a very small increase in $CFC_{mid}$ for increasing $COT_{th}$. This is probably due to fewer clouds leaving the mid-level cloud class than being added from the high level cloud class for increasing $COT_{th}$. Generally, ERA-

Interim $CFC_{mid}$ lies slightly outside the Cloud_cci uncertainty range indicating too few mid-level clouds in ERA-Interim.

For low clouds a good agreement of ERA-Interim $CFC_{low}$ with Cloud_cci is found, the difference being within 10 %. ERA-Interim has fewer clouds than Cloud_cci for nearly all latitudes except in the Arctic regions, where ERA-Interim is about 10 % higher than Cloud_cci, and some parts of the Tropics, where $CFC_{low}$ is nearly equal between both datasets. Looking at the global maps of $CFC_{low}$, the agreement of the spatial patterns between ERA-Interim and Cloud_cci is remarkable. Similar to

mid-level clouds, very little change is found for ERA-Interim $CFC_{low}$ when varying $COT_{th}$, which can have a similar reason as discussed above for mid-level clouds. In addition, low-level clouds are mainly water clouds, which usually have COTs higher than 1.0 anyway.



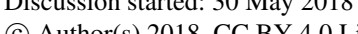


**Figure 4.** Multi-annual mean cloud fraction from ERA-Interim (rows 1-3) and Cloud_cci (row 4) for high (CFC$_{high}$, left column), mid-level (CFC$_{mid}$, middle column) and low clouds (CFC$_{low}$, right column). The ERA-Interim cloud fraction was produced by SIMFERA for three optical thickness thresholds (COT$_{th}$=0.0, 0.15, 1.0; rows 1 to 3, respectively). Bottom row: zonal mean plots with grey shaded areas showing the estimated uncertainty in Cloud_cci CFC$_{high}$, CFC$_{mid}$ and CFC$_{low}$ based on comparisons to CALIOP.





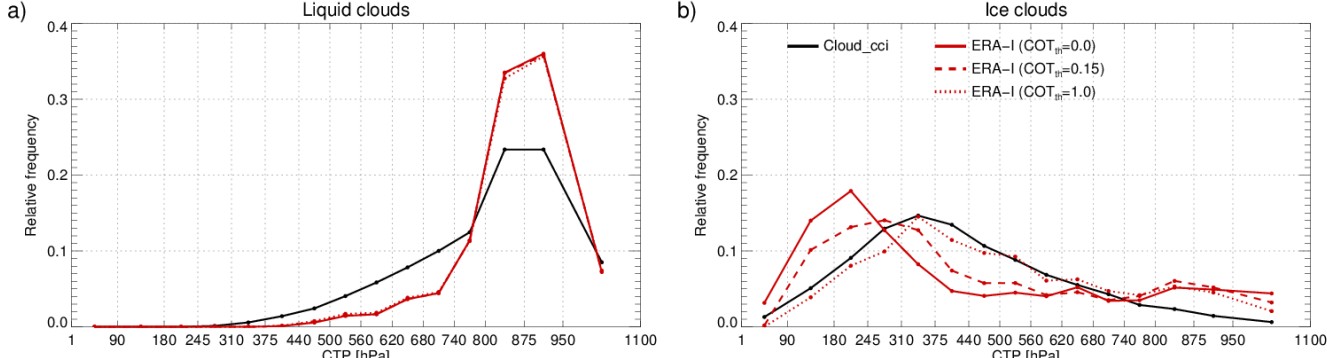

**Figure 5.** Global, multi-annual, relative frequency histograms of observed Cloud_cci cloud top pressure (CTP) compared to ERA-Interim CTP after applying SIMFERA with three COT thresholds (0.0, 0.15 and 1.00) - separated in liquid (left) and ice clouds (right).

Figure 5 reports one-dimensional CTP histograms from ERA-Interim and Cloud_cci, separately for liquid and ice clouds. For liquid clouds there is a negligible impact of the applied COT thresholds on the CTP, as the histograms do not change significantly. This can be explained by the generally high COT of liquid clouds. The agreement between Cloud_cci and ERA-Interim histograms is generally good for liquid clouds, with the histograms of both datasets peaking around 875 hPa. It is
assumed that the histograms would fit even better if the ERA-Interim clouds would be allowed to remain supercooled to lower temperatures, which would add some mid-level liquid clouds to the histograms and at the same time lower the relative frequency of low-level clouds.

When considering ice clouds, applying different COT thresholds for inferring CTP has a significant impact as visible in panel (b) of Figure 5. When applying no threshold ($COT_{th}=0.0$), the ERA-Interim histogram for ice clouds peaks very high
around 200 hPa. This maximum becomes broader and moves to lower levels (higher CTPs) when $COT_{th}=0.15$ and $COT_{th}=1.0$ are applied. For $COT_{th}=1.0$ the ERA-Interim histogram fits the Cloud_cci data quite well. An exception occurs at lower levels, for which, nearly insensitive to the applied threshold, higher ice cloud frequencies are found in ERA-Interim. Further analysis revealed that these low-level ice clouds are located in high latitude regions (not shown). It can be speculated that these low ice clouds exist at relative warm sub-zero temperatures and would disappear from the ice statistics if liquid cloud water were
allowed to remain liquid until lower temperatures in ERA-Interim.



## 4.3   Cloud thermodynamic phase



**Figure 6.** Multi-annual mean cloud phase (CPH), presented as liquid cloud fraction, from ERA-Interim (panels a-c) and Cloud_cci (panel d), where the ERA-Interim liquid cloud fraction was produced by SIMFERA for three top-down optical thickness thresholds (COT$_{th}$=0.0, 0.15, 1.0), at which the phase was collected from ERA-Interim profiles. Panel e: zonal mean plot of CPH for all 4 sets. The grey shaded area corresponds to the estimated uncertainty in Cloud_cci CPH based on comparisons to CALIOP.

Figure 6 shows multi-annual mean CPH for Cloud_cci AVHRR-PM data and for ERA-Interim, the latter processed through SIMFERA for COT$_{th}$ of 0.0, 0.15 and 1.0. A general similarity of the global patterns of ERA-Interim to Cloud_cci is found with, for example, very high CPH in the marine stratocumulus regions, relatively lower values in the inner Tropics and lowest CPH values over Antarctica. However, the differences between ERA-Interim and Cloud_cci are large in the mid and high Latitudes where CPH drops much more in ERA-Interim than in Cloud_cci, meaning that ERA-Interim has much higher relative ice cloud frequencies. The occurrence of too few liquid clouds for sub-zero temperature in ERA-Interim is likely caused by a too strict, linear, liquid-to-ice conversion suppressing all liquid clouds below -23° C. This would be in line with the discussion about





the CTP histograms in the previous paragraph. This issue was also addressed in (Forbes et al., 2016), according to which too few liquid clouds caused too low top-of-the atmosphere upward solar radiation, although their study was limited to cold air outbreaks north of the Antarctic ice shield.

When applying $COT_{th}$=0.15 or $COT_{th}$=1.0 more liquid clouds are sampled (increase in CPH) in ERA-Interim, relatively speaking, which seems natural as lower levels usually have higher temperatures than the levels above. While the increase in ERA-Interim CPH is only moderate for $COT_{th}$=0.15 compared to $COT_{th}$=0 (increase of about 10% CPH in the Tropics and decreasing impact with higher latitudes), the CPH increase is large for $COT_{th}$=1.0 (up to 30 % nearly everywhere). The uncertainty in Cloud_cci (an estimate of which is shown as grey-shaded area in Figure 6e) is rather small and cannot explain the found differences between ERA-Interim and Cloud_cci.

## 5   Summary and conclusions

Global atmospheric models are usually not capable of resolving small-scale, sub-grid processes related to clouds and therefore use prognostic cloud schemes including parametrizations for connecting grid box variables to sub-grid processes. This is not only true for GCMs, but also for atmospheric reanalyses, which employ numerical weather prediction models over multiple decades of data. On the one hand, reanalysis data sets can be considered to be very accurate, e.g. for thermodynamic profiles such as temperature and moisture, due to the cyclic data assimilation conducted. On the other hand, their cloud properties are still modelled exclusively, giving the opportunity to evaluate the cloud-related parametrizations used (which are closely related to those of GCMs) with observations. Satellite observations are the only source that can provide reference observations with global coverage and at spatial scales of clouds. However, to overcome the mismatch in spatial scales and in representativeness between models and satellite observations, satellite simulators are necessary.

In this study a simplified satellite simulator (called SIMFERA) is introduced and applied to ERA-Interim fields with a focus on cloud properties. Like already existing satellite simulators, SIMFERA is used to convert the model state into pseudo satellite observations. Input to SIMFERA are basic model output fields (e.g. temperature, moisture, cloud cover and cloud condensate), complemented by using model parametrizations for those properties no directly available such as effective radius. More complex simulators often use synthetic radiances determined by a radiative transfer model (RTM) using the model fields as input to derive synthetic retrievals of cloud properties. Most commonly they are run online (as part of COSP for example) using the temporarily available model fields such as optical properties. Unlike these, SIMFERA can be run offline on high temporal resolution basic model output, keeping modifications to a minimum and mainly tackling the difference in horizontal and vertical scales and resolutions. This approach enables more flexibility of the simulator and supports a clearer back-propagation of the results of comparisons to observations to actual model deficiencies. In more detail, SIMFERA is build up of three modules: (1) a preprocessor, which determines all necessary input data from available model fields for each grid box; (2) a downscaler, which converts the model grid box mean profiles to sub-grid column profiles considering the sub-grid variability and the finer spatial resolution of the satellite pixel; (3) pseudo-retrieval, which emulates the pixel-scale cloud property retrieval per column followed by statistical aggregation, which is strongly aligned with the content of the observational





dataset (e.g., temporal averages and histograms). In summary, unaveraged model fields, such as available from ERA-Interim reanalysis, undergo the SIMFERA adjustment and are then aggregated to mimic existing satellite-based cloud climatologies such as the Cloud_cci AVHRR-PM dataset. This dataset was then used in subsequent comparison (for the period 1982-2014) acting as reference and including estimates of systematic uncertainties.

The comparisons reveal that in terms of global patterns ERA-Interim total cloud fraction agrees very well to the observations. However, it is biased low nearly everywhere on the globe. This underestimation amounts to approximately 10%. Exceptions are the polar regions in which ERA-Interim has partly much higher cloud fraction than the observations. This is caused by the occurrence of very thin clouds as the overestimation disappears when clouds with low optical thicknesses are filtered out. The analysis revealed further that the underestimation in total cloud fraction is mainly caused by a lack of mid-level and low

clouds, although the spatial patterns agree well in particular for low clouds. High cloud occurrence in ERA-Interim agrees to the observations within their uncertainties. With respect to cloud phase it is found that ERA-Interim has a significant ice bias. This bias in the corresponding liquid cloud fraction amounts partly to 30 % and is most pronounced in the northern mid-latitudes, and north- and south-hemispheric polar regions. It is very likely that this is caused by the suppression of liquid clouds below -23° C in ERA-Interim cloud schemes.

In upcoming versions of the Cloud_cci datasets optical and microphysical cloud properties will be improved, facilitating the evaluation of corresponding modelled equivalents of those as well.

*Competing interests.* The authors declare that no competing interests are present.

*Acknowledgements.* This work was supported by the European Space Agency through the Cloud_cci project (contract No.: 4000109870/13/I-NB).





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
