# Peer review of "Comparing ERA-Interim clouds with satellite observations using a simplified satellite simulator"

_Atmospheric Chemistry and Physics, 2018_

## Referee Comment (RC1) · Anonymous Referee #1 · 21 Jun 2018

This paper describes a new satellite simulator software compatible with the Cloud_cci AVHRR datasets, and presents an initial application to ERA Interim reanalysis. The topic is relevant and the paper is clearly written. However I have two major concerns: the analysis is relatively superficial, limiting its utility, and the description of the simulator lacks detail in some respects. I believe this paper requires a major revision before it can be accepted for publication. Please see more elaborated comments below.

- The abstract, introduction, and conclusions insist that the approach presented here supports a clearer understanding of model deficiencies. This seems to me like a very general statement that is not properly supported by the evaluation presented here. I

[Figure]

think this stems from the fact that the analysis presented here is rather superficial, and therefore it is difficult to see how the model can be modified to address some of the errors discussed in this paper.

- As far as I can see, the CTP calculation is not explained with enough detail, specially how it depends on the COT thresholding. According to L10-11 in Page 10, the COT threshold removes the cloud cells above the level where the COT threshold is reached. However, it is not clear how the cloud top phase is affected by this. How this is done may have significant impacts in multi-layer situations: if a thin, high cloud layer is removed, the algorithm will report a much larger CTP, but in observations the retrieval algorithm is applied to the entire column, introducing an inconsistency in the simulator CTP.

- In recent years, several papers looking at the evaluation of cloud phase in models have been published (e.g. papers by G. Cesana and J. Kay). It would be worth citing some of these papers.

- P8, L8-13. Is this approach consistent with the subcolumn generation algorithm used in IFS?

- As far as I know, at least one study has been published that applies a simulator to ERA-I clouds: https://link.springer.com/article/10.1007/s00382-016-3204-6

- How the specific value of COT=0.15 is chosen? Is it an estimate of the sensitivity of the Cloud_cci retrieval?

---

## Referee Comment (RC2) · Anonymous Referee #2 · 31 Jul 2018

Initial comments

This paper needs a revision before it can be accepted for publication.

Recommendations for the changes:

1) Page 2, line 15. Change "general circulations model" to "general circulation model".

2) Page 3, line 3. Change "a course spatial" to "a coarse spatial".

3) Page 2, line 19, on "...of the past decades very accurately". I do not think the reanalysis datasets have such a merit. As a matter of fact, many published studies reveal there are discontinuities in reanalysis time series in the context of climate trend detection.

4) There are a significant number of published studies comparing long-term changes in cloudiness from different observing systems including weather stations, satellite radiance derived, and NWP output. Those studies were done without using the cloud simulator method but they help understand and solve issues in long-term cloud monitoring. I think the authors should add one paragraph in Introduction discussing this topic. They should cite some papers on that discussion, including:

a) Dai et al. (2006): Recent trends in cloudiness over the United States: A tale of monitoring inadequancies. BAMS, 87, 597-606.

b) Sun et al. (2015): Variability and trends in U.S. cloud ocver: ISCCP, PATMOS-x, and CLARA-A1 compared to homogeneity-adjusted weather observations. J Clim, 4373- 4389.

Of course, the authors are encouraged in their paper to comment/discuss/quantify what impact would be for those long-term time series comparisons that cloud simulator was not utilized.

5) Free et al. (2016): Comparison between total cloud cover in four reanalysis products and cloud measured by visual observations at U.S. weather stations. J Clim, 2015-2021. This paper pointed out that the reanalsis products including ERA-Interim underestimate cloud cover and overestimate downward solar radiation. This paper should be cited in the authors work to support their finding.

6) A suggestion: Do the author compare their products with PATMOS-x that is virtually derived from the same satellite instruments? it maybe a good idea to include the results from that comparison for discussion.

---

## Author Comment (AC2) · 25 Sep 2018

We would like to thank the reviewer for the very valuable comments and suggestions. We believe that the implementations of these have improved the paper significantly.

Authors' response to RC1 - Anonymous Referee #1
(*Reviewer comments in black and italic*; our responses in blue)

*This paper describes a new satellite simulator software compatible with the Cloud_cci AVHRR datasets, and presents an initial application to ERA Interim reanalysis. The topic is relevant and the paper is clearly written. However I have two major concerns: the analysis is relatively superficial, limiting its utility, and the description of the simulator lacks detail in some respects. I believe this paper requires a major revision before it can be accepted for publication. Please see more elaborated comments below.*

*-The abstract, introduction, and conclusions insist that the approach presented here supports a clearer understanding of model deficiencies. This seems to me like a very general statement that is not properly supported by the evaluation presented here. I think this stems from the fact that the analysis presented here is rather superficial, and therefore it is difficult to see how the model can be modified to address some of the errors discussed in this paper.*

Yes, our study investigates the quality of modelled clouds by comparisons to observations. By this we believe some model deficiencies become very clear. We believe this is in particular strengthened by the incorporation of the uncertainties associated with the observations. Some of the model deficiencies that have been clearly demonstrated: (a) the ERA-Interim cloud fraction is biased low for mid-level and low-level clouds. (b) ERA-Interim lacks super-cooled liquid clouds and has too many ice clouds for temperatures between 0 and -40°C.
This information will be very useful for people using ERA-Interim cloud information in other applications. This information is expected to be also helpful for model developers who will probably know what further model developments will have to focus on to reduce/eliminate the model deficiencies found.
Explicit elaboration on which model developments will have to be carried out, based on our findings, is outside the scope of our study.

*- As far as I can see, the CTP calculation is not explained with enough detail, specially how it depends on the COT thresholding. According to L10-11 in Page 10, the COT threshold removes the cloud cells above the level where the COT threshold is reached. However, it is not clear how the cloud top phase is affected by this. How this is done may have significant impacts in multi-layer situations: if a thin, high cloud layer is removed, the algorithm will report a much larger CTP, but in observations the retrieval algorithm is applied to the entire column, introducing an inconsistency in the simulator CTP.*

Yes, it is correct that all cloud top layers are removed from a column until a COT threshold is reached (top-down). After this, the upper-most cloud layer determines the cloud top phase. In case of an optically very thin ice cloud layer above a lower level liquid cloud layer, the cloud top phase will be liquid if the ice cloud layer has a COT that is lower than the COT threshold. It can also cause the resulting CTP for that column to be much larger than the cloud top pressure of the ice cloud layer that is removed. Even though the measurement and retrieval of passive imager data contains rather an integrated signal, the approach above needs to be applied to make model and satellite data comparable due to the limited capabilities of passive imagers in that respect. This approach is also absolutely consistent with the conducted validation exercises (in which

COT thresholds are applied to CALIOP profiles) in which the errors in the satellite data, i.e. the systematic errors, are quantified, which play a crucial role in our study when interpreting the comparison results.

Thus, we believe the presented approach is ensuring consistency rather than introducing inconsistency. But we understand that this needs to be explained better in the text. We will add a paragraph at the end of Section 3.2

*- In recent years, several papers looking at the evaluation of cloud phase in models have been published (e.g. papers by G. Cesana and J. Kay). It would be worth citing some of these papers.*

Thank you for this suggestion: we add references to Cesana et al. (2012, GRL), and Weidle and Wernli (2008, JGR Atmosphere)

*- P8, L8-13. Is this approach consistent with the subcolumn generation algorithm used in IFS?*

We assume the reviewer refers to the Monte-Carlo Independent Column Approximation, (McICA) which was introduced in the ECMWF radiation schemes (McRad, Morcrette et al., 2008a) into ECMWF IFS operational libraries with Cy32r2 in June 2007 (https://www.ecmwf.int/en/elibrary/9211-part-iv-physical-processes). Although our approach has some similarities to the McICA approach, it is more similar to approaches in typical satellite simulators, e.g. of COSP (Bodas-Salcedo et al., 2011). But most importantly, the McICA approach was not contained in IFS Cy31r1, which was used for ERA-Interim, the data we used in our study.

*- As far as I know, at least one study has been published that applies a simulator to ERA-I clouds: https://link.springer.com/article/10.1007/s00382-016-3204-6*

Thank you. We will add this reference to the introduction.

*- How the specific value of COT=0.15 is chosen? Is it an estimate of the sensitivity of the Cloud_cci retrieval?*

The COT threshold 0.15 is indeed chosen with respect to the sensitivity of the Cloud_cci cloud mask and phase detection. This threshold describes the approximate value below which the majority of clouds are missed or the phase misclassified. The value of 0.15 is based on validation studies against CALIOP, as explained in the paragraph on page 5 line 16 to page 6 line 7. Investigating additionally 0.0 and 1.0 as thresholds is meant to provide an estimate of the sensitivity of the chosen approach with respect to the selected COT threshold.

We will make this clearer at the end of Section 3.2.

---

## Author Response (AR2)

========================================================================
Authors' response to Anonymous Referee #1
(*Reviewer comments in black and italic*; our responses in blue)

*Review of Stengel et al., Comparing ERA-Interim clouds with satellite observations using a simplified satellite simulator.*

*Thanks for addressing a couple of points that I thought were unclear in the previous version of the manuscript. The paper is a nice documentation paper of this new software tool, but I still believe that the analysis presented in the is rather superficial. Although I recommend publication, I believe this paper would fit better in the sister journal Geoscientific Model Development, rather than in ACP. I encourage the editor to think about this option.*

Thank you for the acceptance of the manuscript

*The authors don't mention if the code is available for public use. It would be very helpful to add a sentence on this.*

The code is indeed freely available through a github repository (https://github.com/martinstengel/simfera). We have added this information to the 'code availability' section at the end of the manuscript.

[revised manuscript text omitted]